# Integrative and comparative single-cell analysis reveals transcriptomic difference between human tumefactive demyelinating lesion and glioma

Xiao-Yong Chen[1,3], Yue Chen[1,3], Wen-Hua Fang[1,3], Zan-Yi Wu[1], Deng-Liang Wang[1], Ya-Wen Xu[1], Liang-Hong Yu[1], Yuan-Xiang Lin [1✉], De-Zhi Kang [1✉] & Chen-Yu Ding [1,2✉]

Tumefactive demyelinating lesion (TDL) is an immune-mediated disease which can be misdiagnosed as glioma. At present, there is no study comparing difference between the two disorders at the cellular level. Here, we perform integrative and comparative single-cell RNA sequencing (ScRNA-seq) transcriptomic analysis on TDL and glioma lesions. At single-cell resolution, TDL is comprised primarily of immune cells, which is completely different from glioma. The integrated analysis reveals a TDL-specific microglial subset involving in B cell activation and proliferation. Comparative analysis highlights remyelination function of glial cells and demyelination function of T cells in TDL. Subclustering and pseudotime trajectory analysis of T cells in TDL reveal their heterogeneity and diverse functions involving in TDL pathogenesis and recovery process. Our study identifies substantial differences between TDL and glioma at single-cell resolution. The observed heterogeneity and potentially diverse functions of cells in TDL may be critical in disease progression.

[1] Department of Neurosurgery, Neurosurgery Research Institute, The First Affiliated Hospital, Fujian Medical University, Fuzhou, Fujian, China. [2] Jianning County General hospital, Sanming, Fujian, China. [3]These authors contributed equally: Xiao-Yong Chen, Yue Chen, Wen-Hua Fang. ✉email: lyx99070@126.com; kdz99988@vip.sina.com; dingcydr@163.com

Tumefactive demyelinating lesion (TDL) is a rare, focal destructive disorder in the central nervous system (CNS) which appears tumor-like. The histological features of TDL are similar to typical multiple sclerosis (MS). While TDL lesions could occur in inflammatory diseases other than MS[1]. As a solitary mass lesion, it closely mimics a brain tumor clinically, radiologically, and pathologically. Therefore, it is often misdiagnosed as a brain tumor in clinics, especially high-grade glioma. Currently, many studies have reported individual cases of TDL mimicking brain glioma and revealing the difference between them from clinical, radiological, and pathological aspects[2–7]. However, there is no study comparing the difference between the two disorders at the cellular level.

Recently, single-cell RNA sequencing (ScRNA-seq) techniques have been developed and become a powerful tool to analyze the expression of thousands of genes in thousands of cells at single-cell resolution. This emerging technique provides an opportunity to comprehensively define cell types and functional states with molecular precision. So far, ScRNA-seq has been successfully applied to depict cellular expression atlases of many disorders[8–13].

In this study, we presented the ScRNA-seq analysis of TDL together with World Health Organization (WHO) grade III glioma and compared the cell types and cell states between them, which may deepen our insight into TDL. The integrated analysis reveals that TDL is immune dominant and glioma is immune scarce, highlighting a TDL-specific microglial subset involved in B cell activation and proliferation. Comparative analysis reveals the remyelination function of glial cells and the demyelination function of T cells in TDL. Further subclustering and pseudotime trajectory analysis reveal heterogeneity and diverse functions of T cells in TDL involved in TDL pathogenesis and recovery process.

## Results

### ScRNA-seq revealed immune dominance in TDL and immune scarcity in glioma.

We performed ScRNA-seq analysis of a TDL and a WHO grade III glioma by capturing 6577 cells and 6037 cells, respectively (Fig. 1a). The clinical information of the patients was presented in Supplementary Table S1. The appearance, radiological, and pathological manifestation of lesions were presented in Supplementary Fig. S1a–d. After quality control, 5996 cells from TDL and 5559 cells from glioma were included for biological integration analysis, with a median of 1824 genes, 5472 transcripts per cell, and 6.89% mitochondrial genes (Fig. 1a; Supplementary Fig. S1e). We performed dimensionality reduction and partitioned 11,555 cells into 18 clusters via principal component analysis and shown by Uniform Manifold Approximation and Projection (UMAP) plot (Fig. 1b). The heatmap showed top markers of each cluster (Supplementary Fig. S1f), revealing transcriptional differences among different clusters. Based on automatic annotation and confirmation by typical markers (Fig. 1c) and AUCell results (Supplementary Fig. S2a–h), cell type for the different clusters was defined (Fig. 1b). The distinct separation of the two samples in UMAP plot suggested that the two disorders were strikingly different at the cellular level (Fig. 1d). The cells were mainly divided into three types, including T cell (4805 cells, 41.6%), microglial cell (1334 cells, 11.5%), and nonimmune cell (5416 cells, 46.9%). The feature plot and proportion plot illustrated that TDL was dominant in immune cells comparing to glioma (Median proportion of eight immune clusters: 0.068 (0.026–0.143) in TDL vs 0.005 (0.001–0.018) in glioma, $Z = -2.941$, $P = 0.002$, $n = 8$ independent samples; Fig. 1d–g; Supplementary Data 1). After comparing the differential genes of the disorders, the volcano plot revealed

that TDL upregulated immune-related gene (*CCL4*) and down-regulated myelination-related gene (*PLP1*) and tumor-related genes (*S100B* and *CRYAB*), which conformed with characteristics of the two disorders (Fig. 1h; Supplementary Data 2).

### Single-cell transcriptomics highlighted a TDL-specific microglial subset.

There were four clusters annotated as microglial cells, including microglial cell.1(M.1), microglial cell.2 (M.2), microglial cell.3 (M.3), and microglial cell.4 (M.4) (Fig. 2a). The four clusters all expressed genes that were confirmed to be highly expressed in human MS-associated microglia (Fig. 2b)[14]. M.1 was highly expressed in *SPP1*, *PADI2,* and *LPL*, which were identified as specific markers of MS-associated Hu-C8 microglia and MS-associated C12 microglia in the study reported by Masuda T et al.[14]. As *SPP1* is a common marker for osteogenesis and bone formation, gene enrichment analysis revealed that M.1 in our study were involved in osteoblast differentiation (geneID: *SPP1/GPNMB/CCL3*; adjust *P* value: 0.015). More importantly, M.1 were involved in the regulation of lipid transport, leukocyte chemotaxis, positive regulation of inflammatory response, and regulation of neuron death (Fig. 2c). Therefore, M.1 may be associated with demyelination which corresponds to Hu-C8 microglia and C12 microglia in the previous study[14]. The median proportion of cells from TDL and glioma in the four microglial clusters was not statistically different: 0.794 (0.518–0.962) vs 0.206 (0.038–0.482), $Z = -2.021$, $P = 0.057$, $n = 4$ independent samples (Fig. 2d; Supplementary Data 1). In detail, there was heterogeneous distribution of proportion in four clusters shown in Fig. 2e (Supplementary Data 1). In addition to a cluster (M.3) was dominated by cells from glioma (termed glioma-enriched cluster), the other three clusters (M.1, M.2, and M.4) were dominated by cells from TDL, which were termed TDL-enriched clusters (Fig. 2e). There was no differentially expressed gene (DEG) in M.1, M.2, and M.3 between the two disorders, showing that they may be the homeostatic microglial subsets. The expression of activated microglial genes and genes induced during both demyelination and remyelination (Fig. 2f; Supplementary Fig. S3a) and homeostatic microglial genes (Supplementary Fig. S3b) were mainly distributed in M.1, M.2, and M.3, especially in the former two clusters, revealing that they were in the process from homeostatic state to activated state. We investigated the expression of MS susceptibility genes and found that they were highly expressed in TDL-enriched clusters (Supplementary Fig. S3c), revealing they were closely associated with TDL.

The M.4 was nearly exclusively comprised of cells from TDL (98.6%), which may be TDL-specific microglial cells. To further investigate M.4, we performed enrichment analysis on the Top 20 markers. Disease enrichment analysis confirmed that M.4 was closely associated with demyelinating disease (Supplementary Fig. S3d). GO enrichment analysis revealed that this subset played a crucial role in immune cell proliferation and differentiation and was especially closely associated with B cells, such as B-cell activation and proliferation (Fig. 2g). Gene Set Enrichment Analysis (GSEA) analysis confirmed that M.4 could positively regulate B-cell receptor signaling pathway (Supplementary Fig. S3e).

### Comparative analysis of glial cells between TDL and glioma highlighted the remyelination function of glial cells in TDL.

There were 10 clusters annotated as nonimmune cells, including C1, C2, C5, C7, C8, C10, C12, C13, C16, and C17 (Fig. 3a). By utilizing the canonical markers *SOX2* and *PTPRZ1* which were overexpressed in glioma cells[15], C5 (glial cells) and C17 (astrocytes) exhibited low or no expression and were considered as benign clusters (Fig. 3b; Supplementary Fig. S4a). The proportion

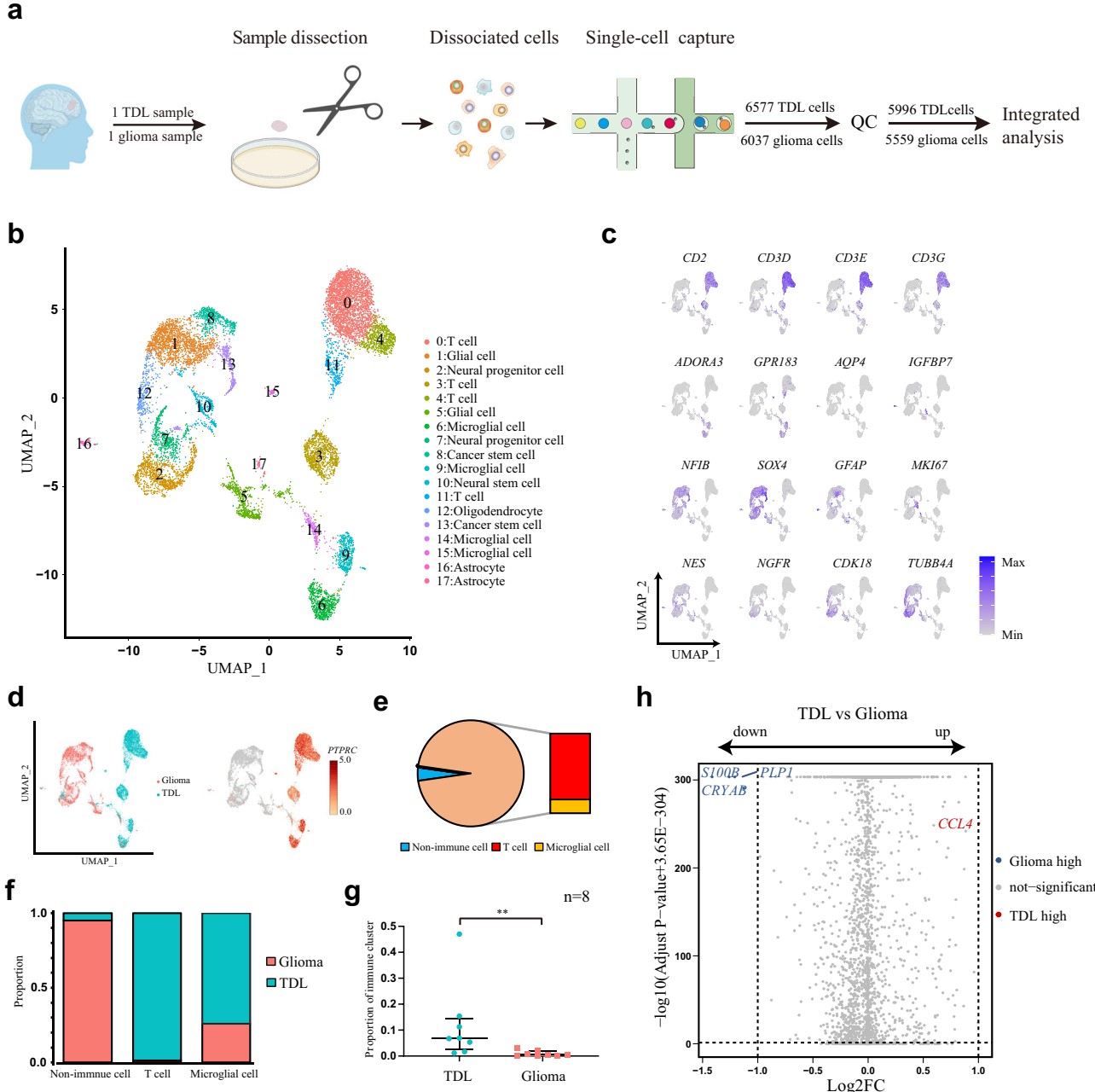

**Fig. 1 The difference of cell-type composition and transcriptomic level between glioma and TDL. a** Workflow diagram (Created with BioRender.com) showing the collection and processing of fresh samples from a high-grade glioma and a TDL for integrated analysis. **b** UMAP plot of cells from the two samples profiled in this study, with each color and number coded to indicate the associated cell types. **c** Feature plots of canonical markers for identify cell types after automatic annotation. **d** UMAP plot indicating the distribution of cells from TDL and glioma with corresponding color (left) and expression level of *PTPRC* in cells (right). **e** Proportion plot of cell-type composition in TDL. **f** Cell-type composition between TDL and glioma. **g** Comparison of proportion of immune clusters in TDL and glioma by nonparametric test (Mann–Whitney test). The data are expressed as median (interquartile range); $n = 8$ independent samples. A *P*-value < 0.05 was considered significant (\**P* < 0.05; \*\**P* < 0.01). **h** Volcano plot of differentially expressed genes (DEGs) that are upregulated (red) or downregulated (blue) in cells of TDL by comparing to cells of glioma. TDL tumefactive demyelinating lesion, QC quality control, UMAP Uniform Manifold Approximation and Projection, FC fold change.

plot revealed that nonimmune cells were mainly derived from glioma, especially in C1, C8, C10, C12, and C13 (Fig. 3c; Supplementary Data 1). The low abundance of astrocytes in TDL may reveal its difference from MS in disease characteristics and pathogenesis[16]. The volcano plot revealed that glial cells in C5 derived from TDL upregulated expression of myelin-related genes (*TYMP, CD9, MAG,* and *PMP22*) and immune-related and inflammatory response-associated genes (*IL1B* and *CXCL8*)

comparing to those from glioma (Fig. 3d; Supplementary Data 3). The downregulated gene (*MAP2*) was mainly associated with tumor progression, which presented the characteristics of glioma-derived glial cells. Enrichment analyses were performed for DEGs between the two disorders in C5. GO enrichment analysis revealed that the DEGs were mainly enriched in receptor-mediated endocytosis, myelination, ensheathment of neurons, axon ensheathment, myelin assembly (Fig. 3e), compact myelin,

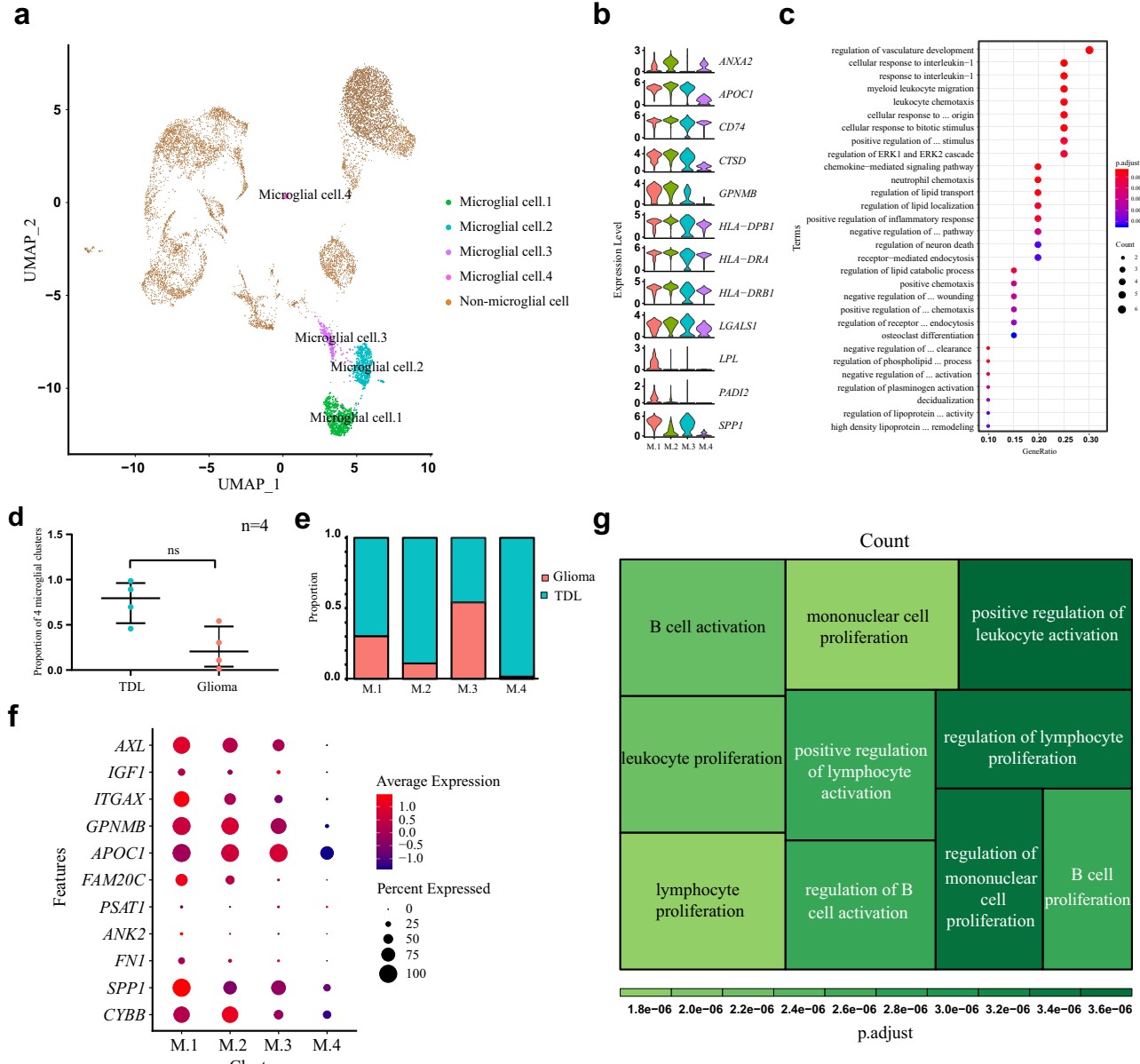

**Fig. 2 Cell clusters of microglial cells and transcriptomic difference between TDL and glioma. a** UMAP plot of microglial cells annotated into microglial cell.1 (M.1), microglial cell.2 (M.2), microglial cell.3 (M.3), microglial cell.4 (M.4). **b** Violin plot of human multiple sclerosis-related genes in four clusters. **c** Bubble plot of Gene Ontology (GO) enrichment analysis (BP biological process) based on top 20 markers in M.1. **d** Comparison of proportion of microglial cells in four clusters from TDL and glioma by nonparametric test (Mann–Whitney test). The data are expressed as median (interquartile range); $n = 4$ independent samples. A $P$-value < 0.05 was considered significant (ns, $P > 0.05$). **e** Proportion plot of microglial cells in four clusters deriving from TDL and glioma. **f** Dotplot of gene expression associated with demyelination and remyelination. **g** Treemap of Gene Ontology (GO) enrichment analysis for biological process based on top 20 markers in M.4. TDL tumefactive demyelinating lesion, UMAP Uniform Manifold Approximation and Projection, FC fold change.

myelin sheath, main axon (Supplementary Fig. S4b), and receptor-ligand activity (Supplementary Fig. S4c). These results suggested that glial cells in TDL may contribute to phagocytic clearance of myelin and remyelination. Similarly, the crucial role of glial cells as promising therapeutic targets has been claimed in demyelinating disease[17,18].

**ScRNA-seq revealed the difference in T cells between TDL and glioma and heterogeneity of T cells in TDL.** The differential abundance of T cells in TDL and glioma has been shown above. After comparing the differential gene expression, the volcano plot (Fig. 4a; Supplementary Data 4) showed that T cells in TDL

upregulated genes associated with immunity (*GZMK, GZMA,* and *CXCL13*) and MS susceptibility gene *RGS1* and downregulated genes associated with myelination (*PLP1*), glial cell differentiation (*GFAP*), and tumor cell proliferation (*SOX2-OT*). Disease enrichment analysis revealed that the DEGs were both enriched in autoimmune disease of the nervous system and central nervous system cancer (Supplementary Fig. S5a). GO enrichment analysis revealed that the downregulated DEGs were mainly enriched in neuron remodeling, myelination, ensheathment of neurons, and axon ensheathment (Supplementary Fig. S5b). These results revealed that T cells in TDL were closely involved in the process of demyelination and its pathogenesis.

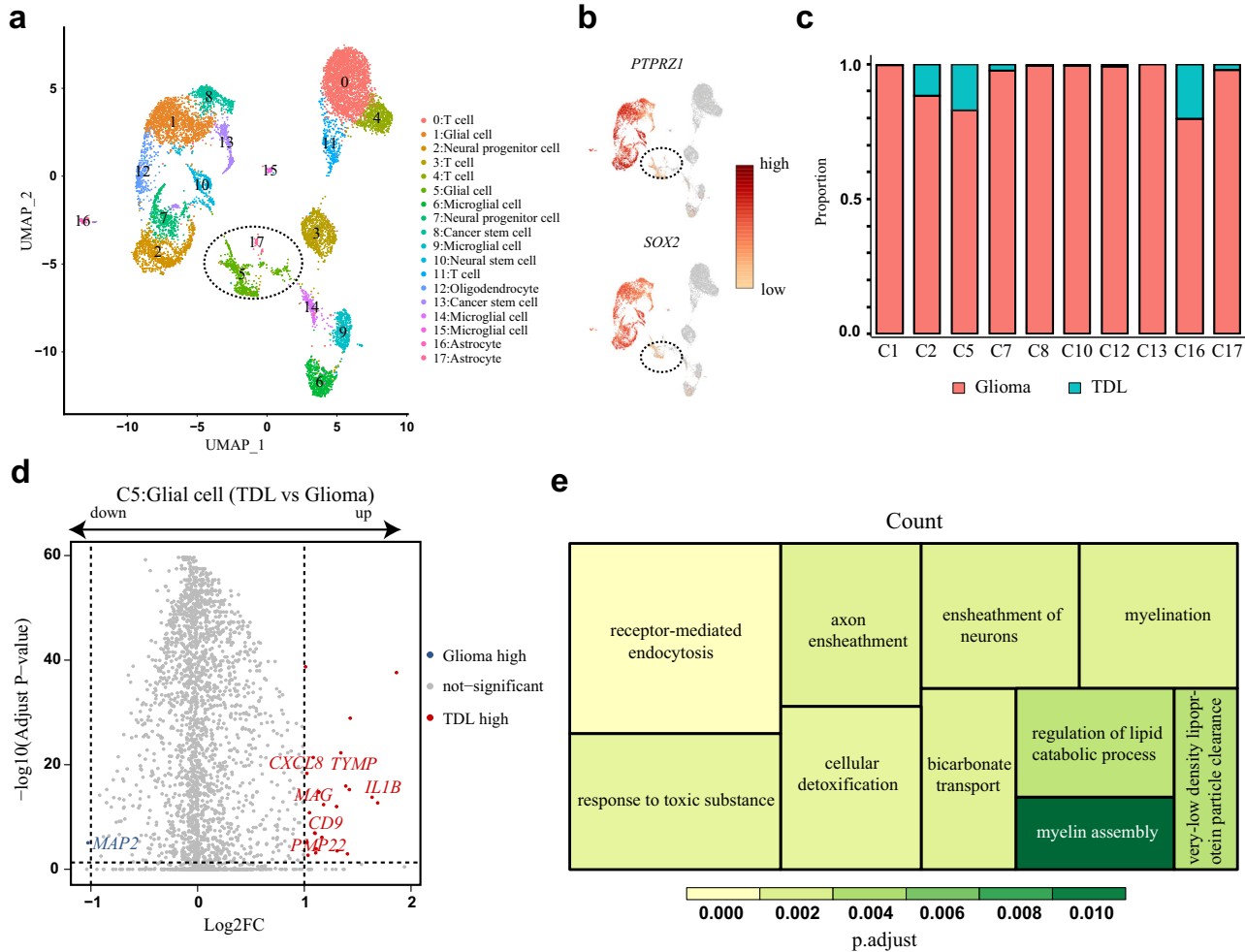

**Fig. 3 Cell clusters of nonimmune cells and transcriptomic difference of glial cells between TDL and glioma. a, b** UMAP plots with expression level of *SOX2* and *PTPRZ1* revealed the two benign clusters (C5, C17) derived from both glioma and TDL. **c** Proportion plot of nonimmune clusters deriving from TDL and glioma. **d** Volcano plot of differentially expressed genes (DEGs) that are upregulated (red) or downregulated (blue) in glial cells (C5) of TDL by comparing to glial cells (C5) of glioma. **e** Treemap of Gene Ontology (GO) enrichment analysis for biological process based on DEGs between C5 cells from TDL and glioma. TDL tumefactive demyelinating lesion, UMAP Uniform Manifold Approximation and Projection. DEGs differentially expressed genes, FC fold change.

To further investigate the heterogeneity of T cells in TDL, a total of 4740 T cells were extracted for reclustering and six subclusters were obtained (Fig. 4b; Supplementary Fig. S5c). Among them, CD4+T cells were the main type, including three subtypes (C0, C4, and C5); a subcluster of CD8+T cells (C3) was identified, and a subcluster characterized by a cytotoxic effector (C1, *IFNG,* and *GZMA*) were identified though they were highly expressed in other clusters (Fig. 4c). As shown in Fig. 4d, for CD4+T cells, C0 was assigned to cytotoxic effector CD4+ T cells, which was characterized by the expression of cytotoxic effectors (*GZMB* and *GZMH*); C4 with high expression of *CCR7, SELL,* and *LEF1* represented naïve CD4+T cells. Meanwhile, C4 showed high expression of cytotoxic effector and effector memory markers (*GNLY, S100A4,* and *ANXA1*), revealing that they may be in activated transition state; C5 was assigned to NK-like CD4+T cells with high expression of *TYROBP*; For CD8+T cells, C3 was assigned to cytotoxic effector CD8+T cells with high expression of *NKG7* and *GNLY*. The expression level of proliferation markers revealed that T cells in TDL were low-proliferative. In addition, C2 covers a respectable proportion of the total T-cell population. However, the cluster is far from other T-cell clusters in UMAP plot after dimensionality reduction, which shows its uniqueness. The cluster has low expression of

*IFNG*, indicating that it appears to play an insignificant role in the innate immune response.

GO enrichment analysis reveals that C2 was mainly involved in very-low-density lipoprotein particle clearance, regulation of cholesterol esterification, high-density lipoprotein particle remodeling, regulation of cholesterol transport, and regulation of axon extension (Fig. 4e). The myelin mainly consists of lipid, proteolipid protein, and cholesterol. Therefore, the main effect of C2 may be involved in remyelination instead of the immune response. In addition, cytotoxic effector CD8+T cells (C3) were associated with T-cell and B-cell activation, myeloid cell homeostasis and development, regulation of leukocyte differentiation and activation, and cell killing (Supplementary Fig. S5d). Top markers of naïve CD4+T cells (C4) were enriched in lymphocyte differentiation and activation, including T cell, B cell, T-helper 1 cell, and alpha-beta T-cell (Supplementary Fig. S5e). Though our study did not identify the existence of T-helper 1 cell and alpha-beta T-cell, the crucial contribution of them have been mentioned in demyelinating disease[19,20]. Intriguingly, NK-like CD4+T cells (C5) were associated with regulation of neuron death and remodeling, neuroinflammatory response, regulation of lipid transport, glial cell activation, macrophage activation, microglial cell activation, lipid transport, high-density lipoprotein particle

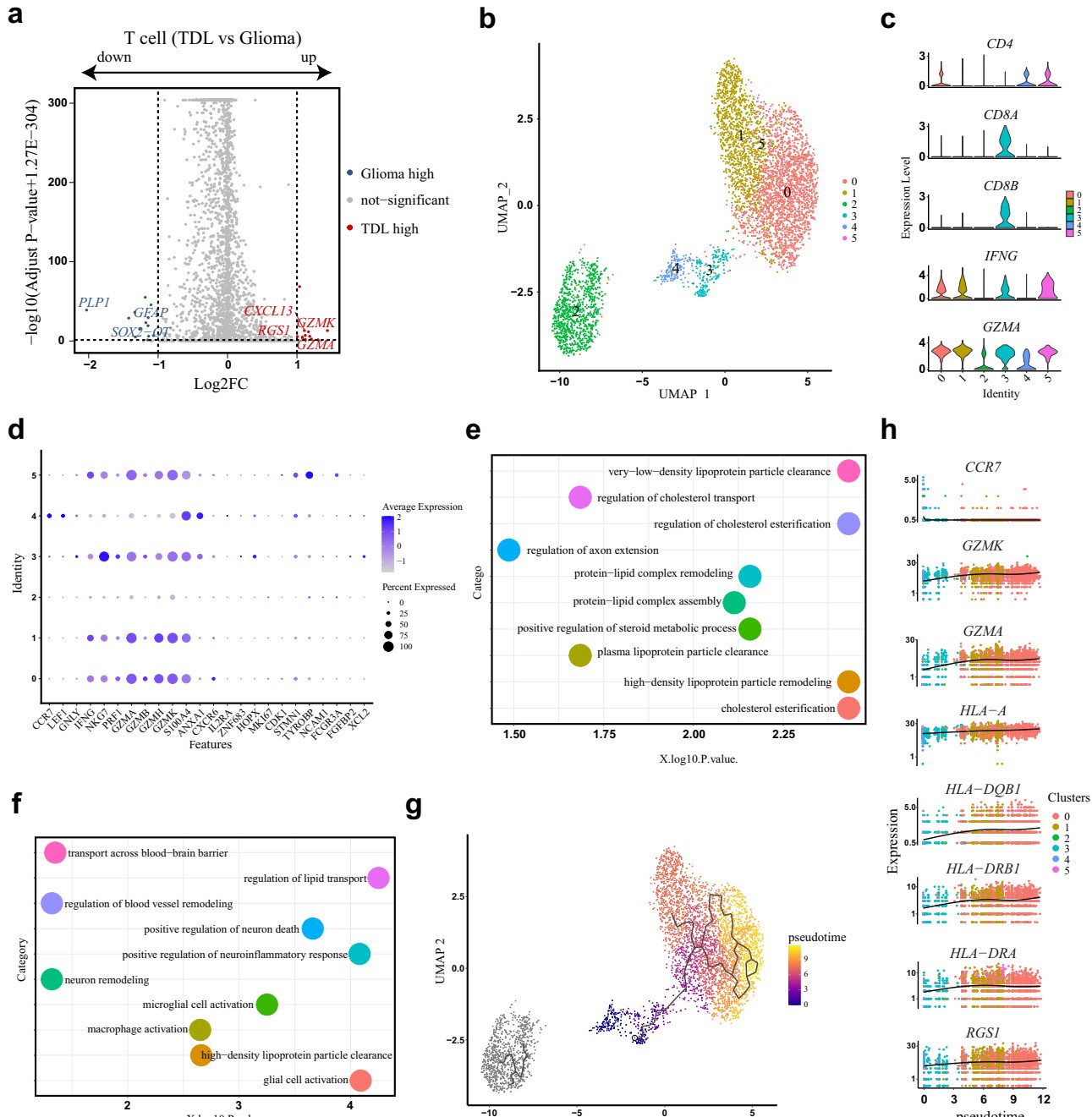

**Fig. 4 Transcriptomic difference of T cells between TDL and glioma and reclustering of T cells in TDL. a** Volcano plot of differentially expressed genes (DEGs) that are upregulated (red) or downregulated (blue) in T cells of TDL by comparing to T cells of glioma. **b** UMAP plot of T cells from TDL after reclustering revealed 6 subclusters (C0, C1, C2, C3, C4, and C5). **c** Violin plots showing the specific expression distribution of selected genes related to T-cell subclusters. **d** Dotplot showing the characteristics of T-cell subclusters. **e** Dotplot of GO analysis (biological process) based on top 20 markers in C2. **f** Dotplot of GO analysis (biological process) based on top 20 markers in C5. **g** The pseudotime trajectory predicted by Monocle 3 and displayed by UMAP plot. Cells were ordered and colored by pseudotime in a gradient from purple to yellow. **h** The dynamic changes of expression level of selected genes along with pseudotime progression. TDL tumefactive demyelinating lesion, DEGs differentially expressed genes, FC fold change, GO Gene Ontology, UMAP Uniform Manifold Approximation and Projection.

clearance, and transport across blood–brain barrier (Fig. 4f). By combing the results above, this subcluster in TDL may be involved in clearing and transporting myelin debris and closely involved in demyelination and remyelination as blood–brain barrier (BBB) breakdown and macrophage infiltration may be the potential mechanism of TDL[21]. In addition, cytotoxic effector T cell (C1) was found to be associated with the maintenance of BBB (Supplementary Fig. S5f).

To further investigate the developmental progression among the T cells, we performed the trajectory analysis via Monocle 3 and presented a putative developmental process of T cells differentiation (Fig. 4g). After considering the biological significance, the initiation node of putative trajectory was selected at naïve CD4+T cells (C4) and cytotoxic effector CD8+T cells (C3). Based on the selected starting point of the developmental stage, the naïve marker *CCR7* was decreased, the cytotoxic effector

(GZMA, GZMK) and MS susceptibility genes (HLA-A, HLA-DQB1, HLA-DRA, HLA-DRB1, and RGS1) were increased along with the increase of pseudotime (Fig. 4h), indicating their crucial role in involving in disease pathogenesis.

## Discussion

TDL may be an independent clinical entity intermediating between MS and acute demyelinating encephalomyelitis. Currently, the etiology of TDL has remained unknown, and there were few studies have been reported. TDL could mimic brain glioma on magnetic resonance images, showing contrast enhancement, mass effect, perilesional edema, and central necrosis[22]. The difficulty of TDL diagnosis often results in surgical resection due to a suspected malignant lesion. Contrast enhancement in TDL was deemed to correlate with BBB breakdown and macrophage infiltration[21]. In addition, contrast enhancement in high-grade glioma could be attributed to BBB breakdown in pre-existing vessels and BBB lack in neovessels. Thus, contrast enhancement in MRI is a common feature of TDL and high-grade glioma, which is ineffective in differentiating them. Contrast-enhancing demyelinating lesions were often misinterpreted as tumors, and numerous cases of the demyelinating disease were initially presumed to be malignant brain tumors due to their large ring-enhancing and space-occupying effect[4,23]. The presented case of TDL in our study was a solitary mass with obvious enhancement on the contrast MRI, resulting in the preoperative diagnosis of high-grade glioma. The histologic features of TDL could also be deceptive, especially those observed in the intraoperative frozen section, resulting in the challenge of differentiating TDL from high-grade glioma[24]. The increased cellularity and pleomorphism in TDL could raise suspicion for infiltrating glioma[3] and perivascular macrophage, and lymphocyte infiltration with myelin loss could contribute to avoid misdiagnosis[3,4]. In addition, the nonspecific symptoms of the two disorders also increase the difficulty in differentiating them. In general, TDL and high-grade glioma share many similar manifestations and are difficult to distinguish.

Our study compared the transcriptomic characteristics of TDL and high-grade glioma and revealed cellular heterogeneity of TDL at the single-cell transcriptome level by scRNA-seq analysis. Though TDL resembles high-grade glioma in many ways, as described above, the results of ScRNA-seq analysis revealed that the two disorders were completely different in cell-type composition and transcriptomic characteristics. In the cell types, TDL was comprised primarily of microglial cells and T cells, while there were few immune cells in high-grade glioma, especially T cells. The overall comparison of transcriptome analysis between TDL and glioma revealed that TDL upregulated immune-related genes and downregulated myelination-related genes and tumor-related genes, presenting the respective characteristics of the two disorders.

The cell-type composition and comparative transcriptome analysis of specific clusters also revealed substantial differences between the disorders. For microglial cells, we identified four clusters, including three homeostatic-to-activated clusters, and one TDL-specific cluster. One of the homeostatic-to-activated cluster (M.1) with high expression of SPP1, PADI2, and LPL may be associated with demyelination which corresponds to Hu-C8 microglia and C12 microglia in the previous study[14]. Enrichment analysis confirmed that the TDL-specific cluster was closely associated with demyelinating disease and played a crucial role in regulating B-cell differentiation and proliferation. Though we did not identify B cells in our samples, a recent study highlighted the importance of B-cell engagement in MS[25], revealing the possibly similar role in TDL. By evaluating the expression of canonical markers of glioma cells, we identified a benign glial cell cluster. The comparative analysis revealed that glial cells from TDL were closely involved in myelination, receptor-mediated endocytosis, myelin assembly, and regulation of the lipid catabolic process. We conjectured that glial cells were mainly responsible for phagocytic clearance of myelin and remyelination in TDL.

Previous studies suggested that demyelinating disease is a T-cell-mediated disorder with immune and inflammation response[26,27]. In our study, we revealed that T cells were the dominant cell type in TDL. The high expression of MS susceptibility gene and low expression of myelination-associated genes revealed their important involvement in the disease initiation and progression. Further, the heterogeneity of T cells in TDL was revealed by extracting and reclustering them. Among them, CD4$^+$ T cells were the main cell subtype. Function enrichment analysis revealed the divergent functional activities in the distinct subclusters. A cytotoxic effector T-cell cluster was associated with the maintenance of BBB, which may be involved in the repair processes. The cytotoxic effector CD8$^+$ T cells exert their function in T-cell activation, regulation of leukocyte differentiation, and antigen binding, suggesting their possible role in disease initiation. In addition, a CD4$^+$ T-cell cluster with NK-like characteristics may be responsible for myelin debris clearance and transport across BBB. Pseudotime trajectory analysis also revealed that MS susceptibility genes increased along with pseudotime progression. In general, the diversity of T cells in TDL revealed their crucial role in disease pathogenesis and the recovery process.

Taken together, our study found the transcriptomic difference between TDL and high-grade glioma at the single-cell transcriptome level. Not only that, we revealed cellular heterogeneity of TDL and identified their specific functions, which may involve in the disease process. These findings deepened our understanding of TDL and may drive future studies on them, especially in TDL.

## Methods

**Patients and samples**. Two adult patients who were diagnosed with WHO III grade glioma or TDL at The First Affiliated Hospital of Fujian Medical University were included in this study. The study was approved by the ethics committee of the First Affiliated Hospital, Fujian Medical University, and conformed to the ethical guidelines of the Declaration of Helsinki. All relevant ethical regulations were followed. Written informed consents of the two patients for specimen collection and further analysis were obtained. The lesion appearance of TDL after surgical excision was shown in Supplementary Fig. S1a. The preoperative and postoperative radiological manifestations of TDL and glioma were shown in Supplementary Fig. S1b. TDL was considered based on several criteria[1]. First, the clinical presentation of the TDL patient is headache, which is a nonspecific symptom suggesting a tumefaction lesion. Second, the lesion is greater than 2 cm with perilesional edema, which is larger than typical MS. The open ring enhancement with an incomplete portion of the ring in MRI is an important diagnostic clue for tumefactive lesions (Supplementary Fig. S1b). Last and most important, the pathological results (Supplementary Fig. S1c–d) reveal the lesion consists of demyelination with hypercellularity and reactive astrocytes containing Creuzfeld cells, which are closely intermingled with myelin-containing foamy macrophages. There are aggregates of numerous T lymphocytes around microvessels. In the follow-up of the patient diagnosed with TDL, the clinical and radiological findings develop in a way supporting our results. In detail, in the patient's follow-up MRIs, the lesion regressed without relapse (Supplementary Fig. S1b); in the clinical follow-up of the patient, there was clinical improvement without any discomfort.

**Tissue processing and single-cell dissociation**. The fresh samples were harvested from the center of the lesion and immediately preserved in tissue preservation solution waiting for further processing. Then, the sample was taken out and washed with phosphate-buffered saline (PBS). After that, the sample was minced into small pieces by scissors on ice and enzymatically digested in brain dissociation kit, which contains 1.25 mL of Enzyme N, 2 × 50 mL of Buffer X (sterile), 1.5 mL of Buffer Y (sterile), 1 vial of Enzyme A (lyophilized powder), and 1 mL of Buffer A. The mixture consisted of sample and enzyme hydrolysate was subjected to the next step for mechanical dissociation via the gentleMACS™ Dissociators. Then, cell clumps and large fragments were strained by a MACS® SmartStrainer (70 μm). After centrifuging cell suspension and resuspending cells with buffer, the final single-cell suspension was generated.

**Library preparation and single-cell sequencing**. According to the manufacturer's protocol, the single-cell suspension was loaded into the Chromium single-cell controller (10X Genomics) for single-cell capturing and downstream library constructions with the Single Cell 3′v2 reagent kit (10x Genomics). Briefly, single cells were suspended and recovered, aiming for 6000 cells. Cells were partitioned into Gel Bead-In-EMulsions (GEMs) with barcoded gel beads. Following cell capture and lysis, complementary DNA was generated and amplified via a S1000TM Touch Thermal Cycler (Bio-Rad). Then, the sequencing libraries were constructed and loaded on an Illumina NovaSeq 6000 sequencer for sequencing.

**ScRNA-seq data processing and identification of cell types**. Raw gene expression matrices were loaded into the statistical computing environment R (4.0.4) and processed by CapitalBio Technology[11,12,28,29] according to the Seurat R package. Three quality criteria were applied to filter the matrices for excluding low-quality cells, including the number of expressed genes (>200), the number of detected transcripts (<99%), and the proportion of mitochondrial genes (≤25%). The gene expression matrices of retained cells were normalized. After that, high variable genes were obtained by "FindVariableGenes" function of Seurat, which served as input genes for dimension reduction by principal components analysis (PCA). UMAP analysis was applied for nonlinear dimensional reduction and visualization of clusters[30]. Based on common features, cells were clustered and projected into two-dimensional space. Clusters were automatically annotated by cellassign[31] and confirmed by expression of canonical markers and enrichment of gene sets by AUCell[32]. Briefly, 8 cell types were identified, including T cell (*CD2*, *CD3D*, *CD3E*, and *CD3G*), glial cell (*GFAP*), neural progenitor cell (*NES* and *NGFR*), microglial cell (*ADORA3* and *GPR183*), cancer stem cell (*SOX4* and *NFIB*), astrocyte (*IGFBP7* and *AQP4*), neural stem cell (*MKI67*), and oligodendrocyte (*CDK18* and *TUBB4A*). We used FindMarkers to identify DEGs among clusters. Log-normalized expression levels of genes were visualized in UMAP plot projections, violin plot, and dotplot.

**Subclustering of T cells in TDL**. For the subtype assessment, T cells in TDL were extracted from the integrated dataset. Then, PCA dimension reduction, clustering, and visualization were performed as described above. Differentially expressed genes among subclusters were also identified via FindMarkers. To define the functional states of T cells, several known genes (*CCR7*, *LEF1*, *GNLY*, *IFNG*, *NKG7*, *PRF1*, *GZMA*, *GZMB*, *GZMH*, *GZMK*, *S100A4*, *ANXA1*, *CXCR6*, *IL2RA*, *ZNF683*, *HOPX*, *MKI67*, *CDK1*, *STMN1*, *TYROBP*, *NCAM1*, *FCGR3A*, *FGFBP2*, and *XCL2*) were used[13].

**Trajectory analysis**. We performed pseudo-ordering of individual T cells by Monocle 3 R package with default parameters. The pseudotemporal dynamics of selected genes were visualized.

**Gene enrichment analysis**. For gene enrichment analysis of each gene set, *P*-value was calculated and adjusted for multiple hypothesis tests via Benjamini and Hochberg method. DEGs was defined as: |log2 fold change| > 1.0, adjusted *p*-value < 0.05, Wilcoxon rank sum test. These enrichment analyses were used to explore the functions of each object, including Disease Ontology (DO), GO (biological process, BP; cellular component, CC; molecular function, MF), KEGG, Reactome, and GSEA.

**Statistics and reproducibility**. Statistical analysis for the sequencing data and their criteria for significance are described above. GraphPad Prism 8 was used for statistical analysis of data other than sequencing data. The proportion of cells from TDL and glioma was compared by a nonparametric test (Mann–Whitney test). Data are expressed as median (interquartile range). The number of samples per group (n) and specific statistical analysis for comparison are described in the corresponding figure legends. A *P*-value < 0.05 was considered significant (**P* < 0.05; ***P* < 0.01).

## Data availability

The original transcriptomic data generated during this study are publicly available in Gene Expression Omnibus (accession ID: GSE202096). Source data for figure panels 1f, 1g, 2d and 3c can be found in Supplementary Data 1.

## Code availability

This study did not generate any unique code. All software tools used in this study are freely available. The authors declare that all R scripts supporting the findings of this study are available from the corresponding author upon reasonable request.

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

## Acknowledgements

This study was supported by grants from Fujian Provincial Health Technology Project (2020GGA051), Special Support Funds for Fujian Province Innovation and Entrepreneurship Talents (2016B010), Technology Platform Construction Project of Fujian Province (2021Y2001), and Technology Platform Construction Project of Fujian Province (2020Y2003). We thank to the guidance of bioinformatics analysis from Suzhou Dynamic Biosystems Co., Ltd.

## Author contributions

X.C., Y.C., and W.F. conceived the study, designed the experiments, and drafted the manuscript. Z.W., D.W., Y.X., and L.Y. collected and prepared specimens. Y.L., D.K., and C.D. supervised this study. All authors have read and approved the final manuscript.

## Competing interests

The authors declare no competing interests

## Consent for publication

All authors have agreed on the contents of the manuscript.

## Ethics approval and consent to participate

The study was approved by the Ethics Committee of The First Affiliated Hospital, Fujian Medical University. All patients provided written informed consent for transcriptomic analysis of their lesions as well as participation in the study.
