## [Peer Review File · Communications Biology]

Reviewers' comments:

Reviewer #1 (Remarks to the Author):

In this study, tissue samples of a patient thought to have TDL and a patient diagnosed with glioma were obtained and single-cell RNA sequencing transcriptomic profiling was examined in lesions. This study is a very valuable and original study in the differentiation of cellular features in the differentiation of TDL and glial tumors. However, I have a few questions regarding the study.

1) It is necessary to explain according to which criteria TDL is considered.

2) In the follow-up of the patient diagnosed with TDL, did the clinical and radiological findings develop in a way that supports the result of the study. Did the lesion regress in the patient's follow-up MRIs? In the clinical follow-up of the patient, was there clinical improvement or did MS develop?

3) It would not be correct to define tumefactive demyelinating lesions as MS variants. TDLs can also occur during other inflammatory diseases, such as acute disseminated encephalomyelitis (ADEM), acute hemorrhagic leukoencephalopathy, NMO, and MS variants (e.g., Schilder's disease, Marburg's disease, and Balo's concentric sclerosis). TDLs have also been reported during viral infection (HIV), malignancy (renal cell carcinoma), autoimmune diseases (e.g., Sjögren's syndrome, Neuro-Behçet's disease, or lupus erythematosus). However, since these diseases are inflammatory in nature, they will not affect the results of the study.

references about this information;

- T. A. Hardy and J. Chataway, "Tumefactive demyelination: An approach to diagnosis and management," *Journal of Neurology, Neurosurgery & Psychiatry*, vol. 84, no. 9, pp. 1047–1053, 2013.

- M. C. Frederick and M. H. Cameron, "Tumefactive Demyelinating Lesions in Multiple Sclerosis and Associated Disorders," *Current Neurology and Neuroscience Reports*, vol. 16, no. 3, 2016.

- M. Abdoli and M. S. Freedman, "Neuro-oncology dilemma: Tumour or tumefactive demyelinating lesion," *Multiple Sclerosis and Related Disorders*, vol. 4, no. 6, pp. 555–566, 2015.

I think that the method and results of the study will make a useful contribution to the differentiation of TDL from glioma. I would like to reevaluate the article with the answers to these questions.

Reviewer #2 (Remarks to the Author):

Chen et al performed integrative single-cell RNA sequencing analysis on Tumefactive demyelinating lesion (TDL) and Glioma patients. In comparison to Glioma, the authors observed enrichment of microglial cells and T cells in TDL patients. The authors next attempted to decode the contributions of the microglial subset from glioma and TDL patients and observed their contributions in B cell activation and proliferation. Further to this, the author's analysis also suggests the remyelination function of glial cells and demyelination of T cells in TDL.

In summary, the authors have addressed a very important problem using scRNA-seq. Please find below my comments that might help authors to further improve their manuscript:

Serious Concerns:

1. While calculating the differentially expressed genes, as shown in the volcano plots, the authors have used some abnormal log₂FC cutoffs. Normally, usage of 2 fold up or down (log scale 1 and -1 respectively) is the standard. However, in Figure 1, the authors have used fold change cutoff (x-axis), lesser than 1 and -1. Similar issues are with other volcano plots throughout the manuscript. This makes the overall data interpretation fishy!

2. Authors have used Monocle 3 for the trajectory inference, and made a statement "The trajectory showed that naïve CD4+ 233 T cells (C4) and cytotoxic effector CD8+ 234 T cells (C3) were at the starting point of developmental stage.". As per my understanding, the initiation node for Monocle 3 is user-defined? How can authors make this statement? This requires further explanation.

Minor Comments:

1. Abstract: I recommend authors spend a considerable amount of time refining the abstract as the present version is highly verbose, and contains multiple grammatical errors.
2. Figure 1: Authors must reconsider adding details about the total number of cells profiled in each case before and after QC.
3. Figure 1: In line 99-100, authors have mentioned that they have used automated cell annotation and marker-based annotation, however, it will be great if they can show some of the bona fide markers of each cell type as a heatmap across all cells. I recommend authors check Seurat output of cell marker depiction.
4. Figure 1: Cluster 8 and 12 are that belong to Glioma are annotated as "other". Please provide the accurate or near accurate annotation of this comparatively large population.
5. Figure 1: Authors have mentioned that "The feature plot and proportion plot illustrated that TDL was dominant in immune cells": However, they have not performed any statistical test (test for proportion) to support their claim (panel D and E).
6. Figure 1F: Authors must add the information in the figure panel to provide direction for the DEG analysis.
7. Figure 2: The required statistical test for proportion is also valid for Panel C of Figure 2.
8. Figure 2: Information about the y-axis is missing in panel B.
9. Figure 2: In panels H and I, the authors must show the NES score as well as its associated FDR values.
10. Figure 3: Volcano plot orientation details are missing in the panel.

Reviewer #3 (Remarks to the Author):

The manuscript entitled "Comparative single-cell analysis reveals the transcriptomic characteristics of MS-variant and tumefactive demyelinating lesion" presents a comparison of disease conditions in TDL and glioma via scRNA sequencing analysis. The authors have extensively studied the microglial and T cell population in depth by clustering them into different subsets for both the diseased states to emphasize their respective roles. Following are some comments for the manuscript:

1. In Figure 4B and 4C the T cell population is clustered. But there is no detailed explanation about the population C2 and its effects. C2 covers a respectable proportion of the total T cell population and have low expression of IFNG (plays a role in innate immune response).
2. In Figure 4a, the authors mention that the volcano plot is showing the gene expressions of T cells

between TDL and Glioma. But it's not very clear which of the genes are correlated to the respective disease condition from the plot.

3. The Y axis of all the volcano plots in the manuscript is not uniform. It would be helpful to keep it uniform throughout.

4. In Figure 2B the expression of specific markers are presented to identify the microglial cell clusters. SPP1 is a common marker for osteogenesis and bone formation. Further references are needed to support its role in MS associated microglia. Additionally, M1 and M4 are termed as TDL enriched clusters but the expression level for SPP1 and LPL are distinctly different for both the clusters, which is overlooked. A possible explanation for these differences is required.

Reviewers' comments:

Reviewer #1 (Remarks to the Author):

In this study, tissue samples of a patient thought to have TDL and a patient diagnosed with glioma were obtained and single-cell RNA sequencing transcriptomic profiling was examined in lesions. This study is a very valuable and original study in the differentiation of cellular features in the differentiation of TDL and glial tumors. However, I have a few questions regarding the study.

1) It is necessary to explain according to which criteria TDL is considered.

2) In the follow-up of the patient diagnosed with TDL, did the clinical and radiological findings develop in a way that supports the result of the study. Did the lesion regress in the patient's follow-up MRIs? In the clinical follow-up of the patient, was there clinical improvement or did MS develop?

3) It would not be correct to define tumefactive demyelinating lesions as MS variants. TDLs can also occur during other inflammatory diseases, such as acute disseminated encephalomyelitis (ADEM), acute hemorrhagic leukoencephalopathy, NMO, and MS variants (e.g., Schilder's disease, Marburg's disease, and Balo's concentric sclerosis). TDLs have also been reported during viral infection (HIV), malignancy (renal cell carcinoma), autoimmune diseases (e.g., Sjögren's syndrome, Neuro-Behçet's disease, or lupus erythematosus). However, since these diseases are inflammatory in nature, they will not affect the results of the study.

references about this information;

- T. A. Hardy and J. Chataway, "Tumefactive demyelination: An approach to diagnosis and management," *Journal of Neurology, Neurosurgery & Psychiatry*, vol. 84, no. 9, pp. 1047–1053, 2013.

- M. C. Frederick and M. H. Cameron, "Tumefactive Demyelinating Lesions in Multiple Sclerosis and Associated Disorders," *Current Neurology and Neuroscience Reports*, vol. 16, no. 3, 2016.

- M. Abdoli and M. S. Freedman, "Neuro-oncology dilemma: Tumour or tumefactive demyelinating lesion," *Multiple Sclerosis and Related Disorders*, vol. 4, no. 6, pp. 555–566, 2015.

I think that the method and results of the study will make a useful contribution to the differentiation of TDL from glioma. I would like to reevaluate the article with the answers to these questions.

Reviewer #2 (Remarks to the Author):

Chen et al performed integrative single-cell RNA sequencing analysis on Tumefactive demyelinating lesion (TDL) and Glioma patients. In comparison to Glioma, the authors observed enrichment of microglial cells and T cells in TDL patients. The authors next attempted to decode the contributions of the microglial subset from glioma and TDL patients and observed their contributions in B cell activation and proliferation. Further to this, the author's analysis also suggests the remyelination function of glial cells and demyelination of T cells in TDL.

In summary, the authors have addressed a very important problem using scRNA-seq. Please find below my comments that might help authors to further improve their manuscript:

Serious Concerns:

1. While calculating the differentially expressed genes, as shown in the volcano plots, the authors have used some abnormal log2FC cutoffs. Normally, usage of 2 fold up or down (log scale 1 and -1 respectively) is the standard. However, in Figure 1, the authors have used fold change cutoff (x-axis), lesser than 1 and -1. Similar issues are with other volcano plots throughout the manuscript. This makes the overall data interpretation fishy!

2. Authors have used Monocle 3 for the trajectory inference, and made a statement "The trajectory showed that naïve CD4+ 233 T cells (C4) and cytotoxic effector CD8+ 234 T cells (C3) were at the starting point of developmental stage.". As per my understanding, the initiation node for Monocle 3 is user-defined? How can authors make this statement? This requires further explanation.

Minor Comments:

1. Abstract: I recommend authors spend a considerable amount of time refining the abstract as the present version is highly verbose, and contains multiple grammatical errors.

2. Figure 1: Authors must reconsider adding details about the total number of cells profiled in each case before and after QC.

3. Figure 1: In line 99-100, authors have mentioned that they have used automated cell annotation and marker-based annotation, however, it will be great if they can show some of the bona fide markers of each cell type as a heatmap across all cells. I recommend authors check Seurat output of cell marker depiction.

4. Figure 1: Cluster 8 and 12 are that belong to Glioma are annotated as "other". Please provide the accurate or near accurate annotation of this comparatively large population.

5. Figure 1: Authors have mentioned that "The feature plot and proportion plot illustrated that TDL was dominant in immune cells": However, they have not performed any statistical test (test for proportion) to support their claim (panel D and E).

6. Figure 1F: Authors must add the information in the figure panel to provide direction for the DEG analysis.

7. Figure 2: The required statistical test for proportion is also valid for Panel C of Figure 2.

8. Figure 2: Information about the y-axis is missing in panel B.

9. Figure 2: In panels H and I, the authors must show the NES score as well as its associated FDR values.

10. Figure 3: Volcano plot orientation details are missing in the panel.

Reviewer #3 (Remarks to the Author):

The manuscript entitled “Comparative single-cell analysis reveals the transcriptomic characteristics of MS-variant and tumefactive demyelinating lesion“ presents a comparison of disease conditions in TDL and glioma via scRNA sequencing analysis. The authors have extensively studied the microglial and T cell population in depth by clustering them into different subsets for both the diseased states to emphasize their respective roles. Following are some comments for the manuscript:

1. In Figure 4B and 4C the T cell population is clustered. But there is no detailed explanation about the population C2 and its effects. C2 covers a respectable proportion of the total T cell population and have low expression of IFNG (plays a role in innate immune response).

2. In Figure 4a, the authors mention that the volcano plot is showing the gene expressions of T cells between TDL and Glioma. But it's not very clear which of the genes are correlated to the respective disease condition from the plot.

3. The Y axis of all the volcano plots in the manuscript is not uniform. It would be helpful to keep it uniform throughout.

4. In Figure 2B the expression of specific markers are presented to identify the microglial cell clusters. SPP1 is a common marker for osteogenesis and bone formation. Further references are needed to support its role in MS associated microglia. Additionally, M1 and M4 are termed as TDL enriched clusters but the expression level for SPP1 and LPL are distinctly different for both the clusters, which is overlooked. A possible explanation for these differences is required.

ONE-BY-ONE RESPONSE TO REVIEWERS' COMMENTS

Reviewers' comments:

Reviewer #1 (Remarks to the Author):

In this study, tissue samples of a patient thought to have TDL and a patient diagnosed with glioma were obtained and single-cell RNA sequencing transcriptomic profiling was examined in lesions. This study is a very valuable and original study in the differentiation of cellular features in the differentiation of TDL and glial tumors. However, I have a few questions regarding the study.

1) It is necessary to explain according to which criteria TDL is considered.

Response: Thank you very much for the suggestion. First, the clinical presentation of the TDL patient is headache, which is a nonspecific symptom suggesting a tumefactive lesion. Second, the lesion is greater than 2 cm with perilesional edema which is larger than typical MS. The open ring enhancement with incomplete portion of the ring in MRI is an important diagnostic clue for tumefactive lesion (Figure S1B). Last and most important, the pathological results (Figure S1C-D) reveal the lesion consists of demyelination with hypercellularity and reactive astrocytes containing Creutzfeldt cells, which are closely intermingled with myelin-containing foamy macrophages. There are aggregates of numerous T lymphocytes around microvessels. Based on these criteria, the patient was diagnosed as TDL. The related reference¹ containing these criteria was cited in our revised manuscript. We have added these descriptions and related figures in our revised manuscript (Location of changes: Lines 337-346; Figure S1B-D).

B

2) In the follow-up of the patient diagnosed with TDL, did the clinical and radiological findings develop in a way that supports the result of the study. Did the lesion regress in the patient's follow-up MRIs? In the clinical follow-up of the patient, was there clinical improvement or did MS develop?

Response: Thank you very much for the comments, it is of great help for revising and improving our paper. In the follow-up of the patient diagnosed with TDL, the clinical and radiological findings develop in a way supporting our results. In the patient's follow-up MRIs, the lesion regress without

relapse (Figure S1B). In the clinical follow-up of the patient, there was clinical improvement without any discomfort. We have added these descriptions and related figures in our revised manuscript (Location of changes: Lines 346-350; Figure S1B).

B

3)It would not be correct to define tumefactive demyelinating lesions as MS variants. TDLs can also occur during other inflammatory diseases, such as acute disseminated encephalomyelitis (ADEM), acute hemorrhagic leukoencephalopathy, NMO, and MS variants (e.g., Schilder’s disease, Marburg’s disease, and Balo’s concentric sclerosis). TDLs have also been reported during viral infection (HIV), malignancy (renal cell carcinoma), autoimmune diseases (e.g., Sjögren’s syndrome, Neuro-Behçet’s disease, or lupus erythematosus). However, since these diseases are inflammatory in nature, they will not affect the results of the study.

references about this information;

- T. A. Hardy and J. Chataway, “Tumefactive demyelination: An approach to diagnosis and management,” *Journal of Neurology, Neurosurgery & Psychiatry*, vol. 84, no. 9, pp. 1047–1053, 2013.

-M. C. Frederick and M. H. Cameron, “Tumefactive Demyelinating Lesions in Multiple Sclerosis and Associated Disorders,” *Current Neurology and Neuroscience Reports*, vol. 16, no. 3, 2016.

-M. Abdoli and M. S. Freedman, “Neuro-oncology dilemma: Tumour or tumefactive demyelinating lesion,” *Multiple Sclerosis and Related Disorders*, vol. 4, no. 6, pp. 555–566, 2015.

Response: We feel great thanks for your professional review work on our article. Based on your nice help and related references, we realized the mistake of the definition about tumefactive demyelinating lesions and revised the related description in our revised manuscript (Location of changes: Lines 1-2, 42-43, 61-63).

I think that the method and results of the study will make a useful contribution to the differentiation of TDL from glioma. I would like to reevaluate the article with the answers to these questions.

Response: Thanks for your nice advice on our article.

Reviewer #2 (Remarks to the Author):

Chen et al performed integrative single-cell RNA sequencing analysis on Tumefactive demyelinating lesion (TDL) and Glioma patients. In comparison to Glioma, the authors observed enrichment of microglial cells and T cells in TDL patients. The authors next attempted to decode the contributions of the microglial subset from glioma and TDL patients and observed their contributions in B cell activation and proliferation. Further to this, the author's analysis also suggests the remyelination function of glial cells and demyelination of T cells in TDL.

In summary, the authors have addressed a very important problem using scRNA-seq. Please find below my comments that might help authors to further improve their manuscript:

Serious Concerns:

1. While calculating the differentially expressed genes, as shown in the volcano plots, the authors have used some abnormal log₂FC cutoffs. Normally, usage of 2 fold up or down (log scale 1 and -1 respectively) is the standard. However, in Figure 1, the authors have used fold change cutoff (x-axis), lesser than 1 and -1. Similar issues are with other volcano plots throughout the manuscript. This makes the overall data interpretation fishy!

Response: Thank you for your valuable comments. In fact, we tried to use 2 fold up or down (log scale 1 and -1 respectively) when calculating the differentially expressed genes (DEGs) as your suggestion. However, based the cutoff, there were little DEGs in our volcano plots for further analysis (Figure 1F: 1 up DEG, 4 down DEGs;; Figure 2F, 0 up and down DEGs; Figure 3D, 24 up DEGs, 1 down DEGs; Figure 4A, 12 up DEGs, 13 Down DEG). Many key genes would not be included in DEGs and the biological differences between the two diseases and specific clusters from the two diseases could not be well explained by DEGs. In addition, we found that the low fold change between clusters may be common in single-cell transcriptomic sequencing analysis. Kim et al² performed integrative single-cell RNA sequencing analysis on Drug-induced hypersensitivity syndrome/drug reaction with eosinophilia and systemic symptoms (DiHS/DRESS) patients and healthy volunteers. In their study, to better display the biological difference between patients and volunteers, the DEGs of specific clusters were defined as follow: DEG: |log fold change| > 0.5, adjusted p value < 0.05, Wilcoxon rank sum test. Therefore, in order to better explain the results of our study and show key genes related to the difference between TDL and glioma under a uniform cuoff, we used their definition of DEGs (log scale 0.5 and -0.5 respectively) to calculate the DEGs as shown in the volcano plots. In addition, we uploaded the original data of volcano plots in our Supplementary Data 1-4 to prove the authenticity and reliability of our results. The related reference was provided and cited in our revised manuscript. Thank you again for your valuable comments.

2. Authors have used Monocle 3 for the trajectory inference, and made a statement "The trajectory showed that naïve CD4+ 233 T cells (C4) and cytotoxic effector CD8+ 234 T cells (C3) were at the starting point of developmental stage.". As per my understanding, the initiation node for Monocle 3 is user-defined? How can authors make this statement? This

requires further explanation.

Response: We feel great thanks for your professional review work on our article. At first, we also had such doubts when we performed pseudo-time trajectory analysis based on Monocle 3. After running Monocle 3, the pseudo-time trajectory could be shown in UMAP plot reflected by black lines among clusters and colors among cells. We found that the black lines representing trajectory in the clusters were fixed and the initiation node was required to be defined by us. Cluster 2 (C2) was far from other clusters in UMAP plot and did not have potential differentiation relationship with other clusters based on trajectory analysis. Therefore, the initiation node could only be C3/C4 or C0/C1/C5. The bioinformatics experts suggested that we could judge and determine the initiation node based on biological significance. As shown in Figure 4H, the expression level of naïve genes was decreased, cytotoxic effect genes and MS susceptibility genes were increased along with pseudo-time trajectory when C3/C4 was selected as initiation node, which is in line with the law of disease development. However, the results would be opposite when we select C0/C1/C5 as initiation node, which would act against the law of disease development. Therefore, we defined the C3/C4 as initiation node based on comprehensive considerations of Monocle 3 results and biological significance. Meanwhile, we also revised the statement into "After considering the biological significance, the initiation node of putative trajectory was selected at naïve CD4+T cells (C4) and cytotoxic effector CD8+T cells (C3). Based on the selected starting point of developmental stage, the naïve marker CCR7 was decreased, the cytotoxic effector (GZMA, GZMK) and MS susceptibility genes (HLA-A, HLA-DQB1, HLA-DRA, HLA-DRB1, RGS1) were increased along with the increase of pseudotime (Figure 4H)" in our latest manuscript. Location of changes: Main text Lines 247-253).

Minor Comments:

1. Abstract: I recommend authors spend a considerable amount of time refining the abstract as the present version is highly verbose, and contains multiple grammatical errors.

Response: Thank you for your nice comments on our article. According to your suggestions, we have refined the abstract (<150 words) and revised our manuscript for grammatical errors under the guidance of professional instructors. Location of changes: Lines 42-55.

2. Figure 1: Authors must reconsider adding details about the total number of cells profiled in each case before and after QC.

Response: Thank you very much for the suggestion. We have added details about the total number of cells profiled in each case before and after QC in Figure 1A and Results section in our revised manuscript. Location of changes: Lines 83-85; Figure 1A.

3. Figure 1: In line 99-100, authors have mentioned that they have used automated cell

annotation and marker-based annotation, however, it will be great if they can show some of the bona fide markers of each cell type as a heatmap across all cells. I recommend authors check Seurat output of cell marker depiction.

Response: Thank you very much for the suggestion, it is of great help for improving quality of our article. As suggested, we have described the bona fide markers (Figure 1C) of each cell type in Method section and showed them as a heatmap (UMAP plot) across all cells based on Seurat output. The related figures (Figure 1C) were uploaded along with our revised manuscript. Location of changes: Lines 386-390; Figure 1C.

4. Figure 1: Cluster 8 and 12 are that belong to Glioma are annotated as "other". Please provide the accurate or near accurate annotation of this comparatively large population.

Response: Thank you very much for the suggestion. The cluster 8, 10, and 12 were annotated as “other” based on automatic annotation tool (Cellassign). Based on expression of canonical markers, we defined cluster 8 as “Cancer stem cell”, cluster 10 as “Neural stem cell”, and cluster 12 as “Oligodendrocyte”. The related figures (Figure 1B, Figure 3A) and descriptions were revised in our latest uploaded manuscript. Location of changes: Lines 386-390; Figure 1B, Figure 3A.

5. Figure 1: Authors have mentioned that "The feature plot and proportion plot illustrated that TDL was dominant in immune cells": However, they have not performed any statistical test (test for proportion) to support their claim (panel D and E).

Response: Thank you very much for the suggestion. As suggested, we performed nonparametric test to compare the proportion of 8 immune clusters in TDL and glioma and found that the proportion of immune cells from 8 clusters in TDL was significantly higher than that in glioma (0.073 vs 0.005, $P = 0.002$; Figure 1G). The statistical results and description were added in our revised manuscript. Location of changes: Lines 97-99; Figure 1G.

G

6. Figure 1F: Authors must add the information in the figure panel to provide direction for the DEG analysis.

Response: Thank you very much for the suggestion, it is of great help for revising and improving our paper. We have added the information in the figure panel (Figure 1H in the latest manuscript) to provide direction for the DEG analysis. Location of changes: Figure 1H.

H

7. Figure 2: The required statistical test for proportion is also valid for Panel C of Figure 2.

Response: Thank you very much for the suggestion. As suggested, we performed nonparametric test to compare the proportion of cells from TDL and glioma in the 4 microglial clusters and did not find difference with statistical significance (0.794 vs 0.223, $P = 0.057$; Figure 2D). In detail, there was heterogeneous distribution of proportion in 4 clusters shown in Figure 2E. The statistical results and description were added in our revised manuscript. Location of changes: Lines 120-123; Figure 2D.

8. Figure 2: Information about the y-axis is missing in panel B.

Response: Thank you very much for the suggestion. As suggested, we have added information about the y-axis in Figure 2B and other violin plots (Supplementary Figure S2A-C).

9. Figure 2: In panels H and I, the authors must show the NES score as well as its associated FDR values.

Response: Thank you very much for the suggestion. As suggested, we have added information of the NES score and its associated FDR values and Nominal p-value in Figure 2 (Figure 2 J-K in the latest manuscript).

10. Figure 3: Volcano plot orientation details are missing in the panel.

Response: Thanks for your careful checks and valuable suggestion. As suggested, we have added the information in the figure panel (Figure 3D) to provide orientation details for the volcano plot.

Reviewer #3 (Remarks to the Author):

The manuscript entitled “Comparative single-cell analysis reveals the transcriptomic characteristics of MS-variant and tumefactive demyelinating lesion“ presents a comparison of disease conditions in TDL and glioma via scRNA sequencing analysis. The authors have extensively studied the microglial and T cell population in depth by clustering them into different subsets for both the diseased states to emphasize their respective roles. Following are some comments for the manuscript:

1. In Figure 4B and 4C the T cell population is clustered. But there is no detailed explanation about the population C2 and its effects. C2 covers a respectable proportion of the total T cell population and have low expression of IFNG (plays a role in innate immune response).

Response: Thank you very much for the suggestion, it is of great help for revising and improving our paper. In our study, the T cell population of TDL is clustered and C2 covers a respectable proportion of the total T cell population. However, the cluster is far from other T cell clusters in UMAP plot after dimensionality reduction and pseudo-time trajectory analysis did not confirm its potential differentiation relationship with other clusters, which shows its uniqueness. In addition, C2 has low expression of IFNG, indicating that it appears to play an insignificant role in innate immune response. Gene enrichment analysis reveals that C2 were mainly involved in very-low-density lipoprotein particle clearance, regulation of cholesterol esterification, high-density lipoprotein particle remodeling, regulation of cholesterol transport, regulation of axon extension (Figure 4E). The myelin is mainly consisted of lipid, proteolipid protein and cholesterol. Therefore, the main effect of C2 may be involved in remyelination instead of immune response. We have added the detailed explanation about the population C2 and its effects with related figure (Figure 4E) in our revised manuscript. Location of changes: Lines 218-228; Figure 4E.

E

2. In Figure 4a, the authors mention that the volcano plot is showing the gene expressions of T cells between TDL and Glioma. But it's not very clear which of the genes are correlated to the respective disease condition from the plot.

Response: We feel great thanks for your professional review work on our article. And we are very sorry for our negligence. We have added the information in the figure panel (Figure 4A) to provide orientation details for the volcano plot and revised the legends of Figure 4A to indicate which of the genes are correlated to the respective disease condition. Location of changes: Lines 681-683; Figure 4A.

A

3. The Y axis of all the volcano plots in the manuscript is not uniform. It would be helpful to keep it uniform throughout.

Response: Thank you very much for the suggestion, it is of great help for revising and improving our paper. In fact, we also noticed that the Y axis of the volcano plots is not uniform. However, the Y axis of the volcano plots is logarithmically transformed from adjust P value of all genes compared between TDL and glioma. And the original adjust P value of different genes in the 4 volcano plots have largely different ranges. In detail, the maximum value of adjust P value was 1 for all the 4 volcano plots, the minimum value of adjust P value corresponding to the 4 volcano plots were extremely different as follows: 0 (Figure 1F), 2.41E-37(Figure 2F), 2.27E-60 (Figure 3D), 0 (Figure

4A). More importantly, we could not directly logarithmically transform the adjusted P value equal to 0. We had to add the second minimum adjusted P value ($3.65E-304$ for Figure 1F; $1.27E-304$ for Figure 4A) to each adjusted P value to visualize all the genes based on \log_2FC and adjusted P value in our volcano plots. Thus, the extremely different ranges of original adjusted P value caused the non-uniform Y axis in 4 volcano plots. In addition, we uploaded the original data of volcano plots in our Supplementary Data 1-4 to prove the authenticity and reliability of our results. Thank you again for your valuable comments.

4. In Figure 2B the expression of specific markers are presented to identify the microglial cell clusters. SPP1 is a common marker for osteogenesis and bone formation. Further references are needed to support its role in MS associated microglia. Additionally, M1 and M4 are termed as TDL enriched clusters but the expression level for SPP1 and LPL are distinctly different for both the clusters, which is overlooked. A possible explanation for these differences is required.

Response: We feel great thanks for your professional review work on our article. And we are very sorry for our negligence. We displayed the expression of specific markers associated with MS in the 4 microglial clusters. Masuda T et al³ revealed context-dependent subtypes of microglia with distinct molecular hallmarks and diverse cellular kinetics in mice with demyelinating and neurodegenerative diseases. In addition, corresponding clusters of microglia were also identified in healthy human brains, and the brains of patients with multiple sclerosis. Based on integration analysis of tissues from healthy human brains and brains of MS patients, two clusters of microglia that were enriched in brains of patients with multiple sclerosis (Hu-C3 and Hu-C8), and one cluster of microglia that was associated with multiple sclerosis (Hu-C2), were clearly separated from the homeostatic clouds on t-SNE plots. Hu-C8 microglia showed strong expression of SPP1, PADI2 and LPL genes, similar to the C12 microglia associated with demyelination in mice. Of note, canonical correlation analysis of mouse and human microglia orthologues confirmed that clusters of microglia (Hu-C2, Hu-C3 and Hu-C8) that are enriched in or associated with the brains of patients with multiple sclerosis have a gene expression profile that is similar to that of clusters of microglia associated with demyelination (C12) and remyelination (C13) in mice. In our study, the markers in Figure 2B were derived from specific markers of Hu-C2, Hu-C3, Hu-C8 in study by Masuda T et al. As you said, expression level for SPP1, PADI2 LPL are distinctly different for the clusters in Figure 2B and are highly expressed in M1. Furthermore, gene enrichment analysis revealed that M1 in our study were mainly involved in regulation of lipid transport, leukocyte chemotaxis, positive regulation of inflammatory response, regulation of neuron death, as well as osteoclast differentiation. Therefore, M1 in our study characterized by high expression of SPP1, PADI2 LPL may be associated with demyelination which correspond to Hu-C8 microglia and C12 microglia in the study reported by Masuda T. We have added the possible explanation based on the reference and its related figure for gene enrichment analysis (Figure 2C) in our revised manuscript. Location of changes: Lines 111-120; Figure 2C.

C

Reference

- 1 Hardy, T. A. & Chataway, J. Tumefactive demyelination: an approach to diagnosis and management. *J Neurol Neurosurg Psychiatry* **84**, 1047-1053, doi:10.1136/jnnp-2012-304498 (2013).
- 2 Kim, D. *et al.* Targeted therapy guided by single-cell transcriptomic analysis in drug-induced hypersensitivity syndrome: a case report. *Nat Med* **26**, 236-243, doi:10.1038/s41591-019-0733-7 (2020).
- 3 Masuda, T. *et al.* Spatial and temporal heterogeneity of mouse and human microglia at single-cell resolution. *Nature* **566**, 388-392, doi:10.1038/s41586-019-0924-x (2019).

Reviewers' comments:

Reviewer #1 (Remarks to the Author):

I think the article is suitable for publication

Reviewer #2 (Remarks to the Author):

While the authors addressed most of my previous concerns, I am still unconvinced with the selection of fold change cutoffs, both for the Volcano plot. For the Volcano plot, I shall recommend authors try and test the NOISeq package in R since it allows computing differential expression between two experimental conditions with no parametric assumptions.

I have concerns about the GSEA, where the authors are reporting the FDR value of 0.9 or 0.25? I recommend the author to recheck since, with the present FDR values, it is hard to accept their proposed results.

=> Figure 2B: Is it expression levels? or Normalized Expression values?

=> By providing the directions in Volcano Plot, what I meant was to indicate whether it is Cluster 1 vs Cluster 2 |or| Cluster 2 vs Cluster 1.

=> Concerning the cellular annotation for the unidentified clusters, I recommend authors to add AUCell results in the suppl data to support their manual cell annotation using the bonafide cell markers.

Reviewer #3 (Remarks to the Author):

The authors have addressed all the comments. It is ready to be published.

Reviewers' comments:

Reviewer #2 (Remarks to the Author):

While the authors addressed most of my previous concerns, I am still unconvinced with the selection of fold change cutoffs, both for the Volcano plot. For the Volcano plot, I shall recommend authors try and test the NOISeq package in R since it allows computing differential expression between two experimental conditions with no parametric assumptions.

I have concerns about the GSEA, where the authors are reporting the FDR value of 0.9 or 0.25? I recommend the author to recheck since, with the present FDR values, it is hard to accept their proposed results.

=> Figure 2B: Is it expression levels? or Normalized Expression values?

=> By providing the directions in Volcano Plot, what I meant was to indicate whether it is Cluster 1 vs Cluster 2 |or| Cluster 2 vs Cluster 1.

=> Concerning the cellular annotation for the unidentified clusters, I recommend authors to add AUCell results in the suppl data to support their manual cell annotation using the bonafide cell markers.

ONE-BY-ONE RESPONSE TO REVIEWERS' COMMENTS

Reviewers' comments:

Reviewer #2 (Remarks to the Author):

1. While the authors addressed most of my previous concerns, I am still unconvinced with the selection of fold change cutoffs, both for the Volcano plot. For the Volcano plot, I shall recommend authors try and test the NOISeq package in R since it allows computing differential expression between two experimental conditions with no parametric assumptions.

Response: We feel great thanks for your suggestions on our article. Based on your suggestions, we try and test the NOISeq package in R for computing differential expression between cells from TDL and glioma. However, the results seem abnormal for the comparison of differential expression in four groups with the DEGs defined as follows: $|\log_2 \text{fold change}| > 1.0$, probability > 0.8 . In detail, we found most genes were upregulated in TDL by comparing to glioma while none gene was upregulated in glioma by comparing to TDL (1. all cells in TDL vs glioma: 2118 up DEGs, 0 down DEGs; 2. C14 cells in TDL vs glioma: 1483 up DEGs, 0 down DEGs; 3. C5 cells in TDL vs glioma: 1733 up DEGs, 0 down DEGs; 4. T cells in TDL vs glioma: 1834 up DEGs, 0 down DEGs). The volcano plots of the NOISeq results were showed below (Volcano plots by NOISeq analysis). After consulting bioinformatics experts, we think these results may not reflect the real biological difference between the two disorders. The reason may be that the NOISeq is not entirely suited to single-cell sequencing data. We have recomputed and determined the accuracy of former results of differential expression analysis. Based on your suggestion, we revised the definition of DEGs as follows: $|\log_2 \text{fold change}| > 1.0$, adjusted p-value < 0.05 , Wilcoxon rank sum test. The

corresponding volcano plots (Figure 1H; Figure 3D; Figure 4A) and results of enrichment analysis (Figure 3E; supplementary Figure 4B-C; supplementary Figure 5A-B) based on DEGs were also revised. Though the cutoffs were changed, the main findings and core concepts of our study were still supported by our latest results. (Location of changes: Lines 93-94,116-117, 145-148, 151-154, 166-170; Figure 1H; Figure 3D-E; Figure 4A; supplementary Figure 4B-C; supplementary Figure 5A-B).

Volcano plots by NOISEq analysis

Figure 1H

Figure 3D-E

Figure 4A

supplementary Figure 4B-C

supplementary Figure 5A-B

A

B

2. I have concerns about the GSEA, where the authors are reporting the FDR value of 0.9 or 0.25? I recommend the author to recheck since, with the present FDR values, it is hard to accept their proposed results.

Response: We feel great thanks for your professional review work on our article. Based on your nice help, we realized the mistaken report of the FDR value. After rechecking the raw value, we found that the FDR values of “B cell receptor signaling pathway” and “B cell proliferation” are 0.04 and 0.90, respectively. Therefore, we retained the GSEA result of “B cell receptor signaling pathway” (supplementary Figure 3E) and deleted that of “B cell proliferation”. Though the high FDR value of GSEA did not support the enrichment of “B cell proliferation”, GO enrichment analysis revealed the significant enrichment of “B cell proliferation” (Figure 2G). Therefore, after combining the results of GSEA and GO enrichment analysis, we proposed that the microglial cluster 4 played a crucial role in B cell proliferation and regulating B cell receptor signaling pathway. (Location of changes: Lines 131-133; Figure 2G; supplementary Figure 3E).

Figure 2G

G

supplementary Figure 3E

E

3. => Figure 2B: Is it expression levels? or Normalized Expression values?

Response: Thank you for your comments on our article. The values of Figure 2B are normalized expression values.

4. => By providing the directions in Volcano Plot, what I meant was to indicate whether it is Cluster 1 vs Cluster 2 |or| Cluster 2 vs Cluster 1.

Response: Thank you for your nice help on our article. In our study, the volcano plots were utilized to show the differentially expressed genes between cells from TDL and glioma. Therefore, we add the directions in the three volcano plots as follows: (1) **TDL vs Glioma** (Figure 1H represent the differentially expressed genes between all cells from TDL and glioma); (2) **C5: Glial cell (TDL vs Glioma)** (Figure 3D represent the differentially expressed genes between C5 cells from TDL and glioma); (3) **T cell (TDL vs Glioma)** (Figure 4A represent the differentially expressed genes between T cells from TDL and glioma). (Location of changes: Figure 1H; Figure 3D; Figure 4A).

Figure 1H

H

Figure 3D

Figure 4A

5. => Concerning the cellular annotation for the unidentified clusters, I recommend authors to add AUCell results in the suppl data to support their manual cell annotation using the bonafide cell markers.

Response: Thank you very much for the suggestion. Based on your suggestion, we have added the AUCell results in the supplementary information to support our manual cell annotation using the bonafide cell markers. (Location of changes: Supplementary Figure S2).

Supplementary Figure S2